# The effect of early pregnancy ALT elevation on neonatal birth weight: The mediating role of gestational diabetes mellitus

**Wen-Xia Ma**[1☯], **Zhou Xu**[2☯], **Rui Xiao**[2], **Xiao-Jun Tang**[1], **Li-Juan Fu**[1,3], **Yan-Xiao Xiang**[1], **Shao-Min Yu**[4], **Yu-Bin Ding**[1]*, **Zhao-Hui Zhong**[1]*

1 Department of Obstetrics and Gynecology, Women and Children's Hospital of Chongqing Medical University, Joint International Research Laboratory of Reproduction and Development of the Ministry of Education of China, School of Public Health, Chongqing Medical University, Chongqing, China, 2 Department of Obstetrics and Gynecology, Sichuan Jinxin Xinan Women and Children's Hospital, Chengdu, Sichuan, China, 3 Department of Pharmacology, Academician Workstation, Changsha Medical University, Changsha, China, 4 Department of Obstetrics and Gynecology, the People's Hospital of Yubei District of Chongqing, Chongqing, China

☯ These authors contributed equally to this work.
* 100144@cqmu.edu.cn (ZHZ); dingyb@cqmu.edu.cn (YBD)

## Abstract

Elevated serum alanine aminotransferase (ALT) levels in early pregnancy and gestational diabetes mellitus (GDM) are linked to an increased rate of large for gestational age (LGA) births. Additionally, elevated ALT levels raise the risk of developing GDM, but it remains unclear whether GDM mediates the effect of ALT on neonatal birth weight. This study examines whether GDM mediates this relationship. We conducted a retrospective cohort study with participants from Jinxin Women's and Children's Hospital who delivered single live births between 2020 and 2023. A multifactorial logistic regression model assessed the relationship between early pregnancy ALT levels, GDM incidence, and LGA births. A mediation model evaluated GDM's role in the impact of elevated ALT on neonatal birth weight. Our study included 12,057 patients. After adjusting for confounders, the difference in LGA rates between elevated and normal ALT groups was significant (OR: 1.248, 95% CI: 1.001–1.556, P=0.049). The GDM incidence difference between these groups was also significant (OR: 1.564, 95% CI: 1.306–1.873, P<0.01), as was the LGA incidence difference between GDM and non-GDM groups (OR: 1.306, 95% CI: 1.129–1.511, P<0.01). After adjusting for confounders, we found that elevated ALT levels in early pregnancy and GDM both affected neonatal birth weight. Specifically, elevated ALT levels had a direct impact on neonatal birth weight (β=0.0291, 95% CI: 0.0100–0.0635), while GDM had an indirect effect (β=0.0025, 95% CI: 0.0012–0.0056), with GDM accounting for 8.1% of the mediation effect. Our study shows that GDM partly mediates the effect of elevated ALT on neonatal birth weight, highlighting the importance of early ALT and glucose screening in routine prenatal care. Healthcare providers should

**Data availability statement:** The underlying data for this study are restricted for ethical reasons by the Ethics Committee of Chongqing Medical University. The dataset includes sensitive medical information and detailed maternal and neonatal clinical records, such as alanine aminotransferase (ALT) levels, gestational diabetes mellitus (GDM) status, and neonatal birth weight. Even after de-identification, there is a potential risk of indirect identification of participants due to the longitudinal nature of the pregnancy-related data and the specific combination of clinical variables. The data are available upon reasonable request from the corresponding author or the institutional data access authority (email: pinyi.chen@cqmu.edu.cn; Phone: +86 023 68485111; Fax: +86 023 63846904).

**Funding:** This work was supported by the National Key Research and Development Program of China (no: 2023YFC2705900) and the Open Fund of Chongqing Maternal and Child Disease Control and Public Health Research Center (no: CQFYSJ01001).

**Competing interests:** The authors have declared that no competing interests exist.

consider including ALT testing in pregnancy protocols and focus on blood glucose control in patients with elevated ALT to reduce the risk of LGA births.

## Introduction

According to the guidelines of the American College of Obstetricians and Gynecologists (ACOG), a newborn whose birth weight is greater than the 90th percentile for gestational age is classified as large for gestational age (LGA) [1]. LGA is associated with various adverse neonatal outcomes. Short-term effects include an increased risk of neonatal admission to the intensive care unit, respiratory distress, metabolic abnormalities, birth injuries, stillbirth, and neonatal mortality. Long-term effects involve a higher risk of developing overweight or obesity, diabetes, cardiovascular diseases, and childhood cancers [2]. Therefore, it is important to effectively reduce the incidence of LGA births by addressing influencing factors.

Research has found that elevated serum alanine aminotransferase (ALT) levels in early pregnancy are associated with an increased likelihood of LGA births in studies examining factors that influence neonatal birth weight [3,4], ALT is a key indicator of liver function during pregnancy, and its elevated levels are a sign of hepatocellular liver damage or potential liver disease. Research shows that even in relatively young and healthy populations, the incidence of abnormal liver tests during pregnancy is about 3% -5% [5]. A prospective cohort study in China showed that the incidence of ALT elevation in pregnant women during early pregnancy reached 24.11% [6]. This emphasizes the clinical importance of monitoring ALT levels as part of routine prenatal care.

Gestational diabetes mellitus (GDM) is another established risk factor for LGA [7]. Compared to non GDM, GDM increases the risk of adverse pregnancy outcomes, and managing blood sugar during pregnancy often improves these outcomes, especially for LGA and macrosomia [8]. A study that included 156 research articles and 7,506,061 pregnant women indicated that after adjusting for confounding factors, women with GDM had a higher likelihood of giving birth to LGA infants (OR: 1.61, 95% CI: 1.09–2.37) compared to women without GDM [9]. GDM can contribute to fetal overgrowth by increasing fetal insulin levels, which promotes excessive growth and adiposity, leading to LGA.

Research has shown that elevated ALT levels in early pregnancy are also associated with the occurrence of GDM [10]. A prospective cohort study showed that for every unit increase in log transformed ALT, the probability of non obese individuals (OR: 3.15, 95% CI: 1.04–9.54) developing GDM increased threefold [3]. A study by Seung Mi Lee and colleagues involving 2,322 women found that the incidence of GDM was 6.5% in patients with elevated ALT levels, compared to 2.1% in the normal ALT group (P < 0.01) [11]. This relationship persisted after adjusting for confounding factors such as maternal age and pre-pregnancy weight, suggesting that ALT elevation could be an early indicator of GDM risk.

In summary, although there is evidence that elevated ALT levels in early pregnancy and the presence of GDM are associated with an increased rate of LGA

births, and that elevated ALT levels increase the risk of developing GDM, it remains unclear whether GDM mediates the effect of ALT on neonatal birth weight. Therefore, this study aims to investigate whether GDM is a mediator in the relationship between elevated ALT levels in early pregnancy and neonatal birth weight. By addressing this gap, we hope to provide further insight into the complex interplay between these factors and inform future strategies for reducing the risk of LGA births.

## Materials and methods

### Study design

A retrospective cohort study was conducted, selecting subjects who underwent prenatal examinations and delivered at Jinxin Women's and Children's Hospital from 2020 to 2023.

Inclusion Criteria: single live birth patients; patients who underwent liver function tests in early pregnancy (11–13 weeks); patients who underwent a 75g oral glucose tolerance test (OGTT) between 24–28 weeks of pregnancy.

Exclusion criteria: patients with unknown gestational age or birth weight; patients with multiple pregnancies; patients with viral hepatitis.

For research purposes, we accessed the data on May 20, 2024. During or after data collection, we were not able to obtain information that would identify individual participants.

### Ethics statement

The study obtained approval from The Ethics Committee of Chongqing Medical University (Approval No. 2022132). All patient data were anonymized to protect privacy and confidentiality. The ethics committees waived the requirement for written informed consent due to the minimal risk associated with using de-identified data.

### Sample size calculation

**Logistic regression analysis.** To detect the significant association between ALT and LGA, the sample size can be calculated using the following formula:

$$N = \frac{(Z_{\alpha/2} + Z_{\beta}) \times [p_1(1-P_1) + P_2(1-P_2)]}{(p_2 - p_1)^2}$$

A literature describing the prevalence of Asian macrosomia and LGA included 47 studies describing the prevalence of LGA in China, and found that the prevalence of LGA in the general population in China is greater than 10% [12]. Assuming $P_1$ (the incidence of LGA in the control group) is 0.10.

Another study exploring the relationship between early pregnancy ALT elevation and LGA suggests that unexplained ALT elevation in early pregnancy is associated with a 4.03-fold increased risk of LGA [3]. Namely OR=4.03, estimate $P_2$ (the incidence of LGA in the exposed group) using the formula $P_2 = P_1 \times OR/(1 - P_1 + P_1 \times OR)$.

Assuming $Z_{\alpha/2}$ (Z value at significance level of 0.05) is 1.96.

Assuming $Z_{\beta}$ (Z value at 90% efficacy) is approximately 1.28.

Substituting the above formula, N is approximately 79.

Considering a 10% dropout rate, the final required sample size is at least 88.

**Analysis of intermediary effect.** For the calculation of sample size in mediation analysis, reference can be made to Fritz and MacKinnon's suggestions [13], and the formula is as follows:

$$n = \frac{8}{f^2} + k + 1$$

$f^2$: The effect size represents the magnitude of the mediating effect on the outcome. Assuming the expected mediating effect size is small, $f^2$ is 0.02.

k: The number of covariates included in the regression model is 11.

Substituting the above formula yields a sample size of approximately 412.

Combining the results of the two analyses, the minimum sample size required for this study is 412. In practice, we included 12,057 research subjects, which fully satisfies the sample size requirement.

## Variable definitions

**Exposure variables.** ALT levels were measured using an enzymatic assay (ALT kit, Roche Diagnostics, USA), with a lower detection limit of 5 U/L and a coefficient of variability of 3.2%. ALT levels were categorized according to the recommended standards for normal adults [14], (with literature indicating no significant difference in ALT levels between early pregnancy and non-pregnant women [15]). This resulted in two groups: the exposed group (ALT > 40 U/L, abnormal) and the non-exposed group (ALT ≤ 40 U/L, normal).

**Outcome variable.** Neonatal birth weight (LGA: infants whose birth weight is above the 90th percentile for gestational age).

**Mediating variables.** Glucose levels were measured using the glucose oxidase-peroxidase method (Beckman Coulter, USA), with a coefficient of variability of 2.8%. All patients were screened for GDM in strict accordance with the Diagnosis and Treatment Guidelines for Gestational Hyperglycemia (2022). At 24–28 weeks of pregnancy, a 75 g oral glucose tolerance test (OGTT) was performed: the blood glucose thresholds on an empty stomach and 1 hour and 2 hours after oral glucose were 5.1, 10.0, and 8.5 mmol/L, respectively. If the blood glucose value reached or exceeded the specified criteria at any time point, it was diagnosed as GDM.

**Covariates.** Maternal age [16,17], pre-pregnancy BMI [18,19], weight gain during pregnancy [20], whether it is the first pregnancy [21], whether to use assisted reproductive technology [22], delivery gestational week, presence or absence of thyroid diseases [23] (including hyperthyroidism, hypothyroidism, and thyroid nodules), presence or absence of gestational hypertension [24] (diagnosed when systolic blood pressure ≥ 140 mmHg or diastolic blood pressure ≥ 90 mmHg, with two blood pressure measurements taken at least 4 hours apart, and postpartum blood pressure returning to normal), presence or absence of pre-eclampsia [25] (diagnosed after 20 weeks of gestation when systolic blood pressure ≥ 140 mmHg and/or diastolic blood pressure ≥ 90 mmHg, with two blood pressure measurements taken at least 4 hours apart, and a urine protein/creatinine ratio (UPCR) ≥ 0.3 or 24-hour urine protein exceeding 300 mg), presence or absence of pre-pregnancy diabetes [26], and presence or absence of intrahepatic cholestasis of pregnancy [27] (diagnosed when bile acids exceed 10 mmol/L).

## Statistical analysis

Appropriate statistical analysis methods were selected based on the type of data. Continuous variables were assessed for normality using the Shapiro-Wilk test. Non-normally distributed variables were presented as medians with interquartile ranges (IQR) and compared using the Mann-Whitney U test, while normally distributed variables were expressed as means with standard deviations (SD) and compared using Student's t-test. Categorical variables were compared using the Chi-Square Test. A multifactorial logistic regression model examined the relationship between early pregnancy ALT levels, GDM, and neonatal birth weight. Hypothesis testing was conducted to ensure that the assumptions of the regression model were met. Mediation analysis was performed using a causal steps approach to estimate the total, direct, and indirect effects of ALT on LGA mediated by GDM. A non-parametric bootstrapping method with 5,000 resamples was applied to derive BCa confidence intervals for indirect effects. This method is widely recognized for its ability to handle non-normal data distributions and is recommended in mediation analysis [28–30]. All analyses were conducted in R using the mediation package. A P value of < 0.05 was considered statistically significant.

## Research process

The overview of the research process is shown below (Fig 1).

## Results

### Baseline data

This study included a total of 12,057 patients with single live births, among whom 1,274 were classified as LGA. The differences in age, BMI, pregnancy weight gain, nulliparous, assisted reproduction, pregnancy hypertension, GDM, pre-pregnancy diabetes, delivery gestational week, and ALT levels between the LGA group and the non-LGA group were statistically significant ($P < 0.05$). The median values of age and delivery gestational week are the same between the two groups, but the difference between the groups is significant. This may be due to the large sample size and the different distribution of these two variables between the groups [31]. However, there were no statistically significant differences in pre-eclampsia, intrahepatic cholestasis of pregnancy, or thyroid diseases between the two groups (Table 1). Shapiro Wilk normality test for continuous variables in S1 Tables.

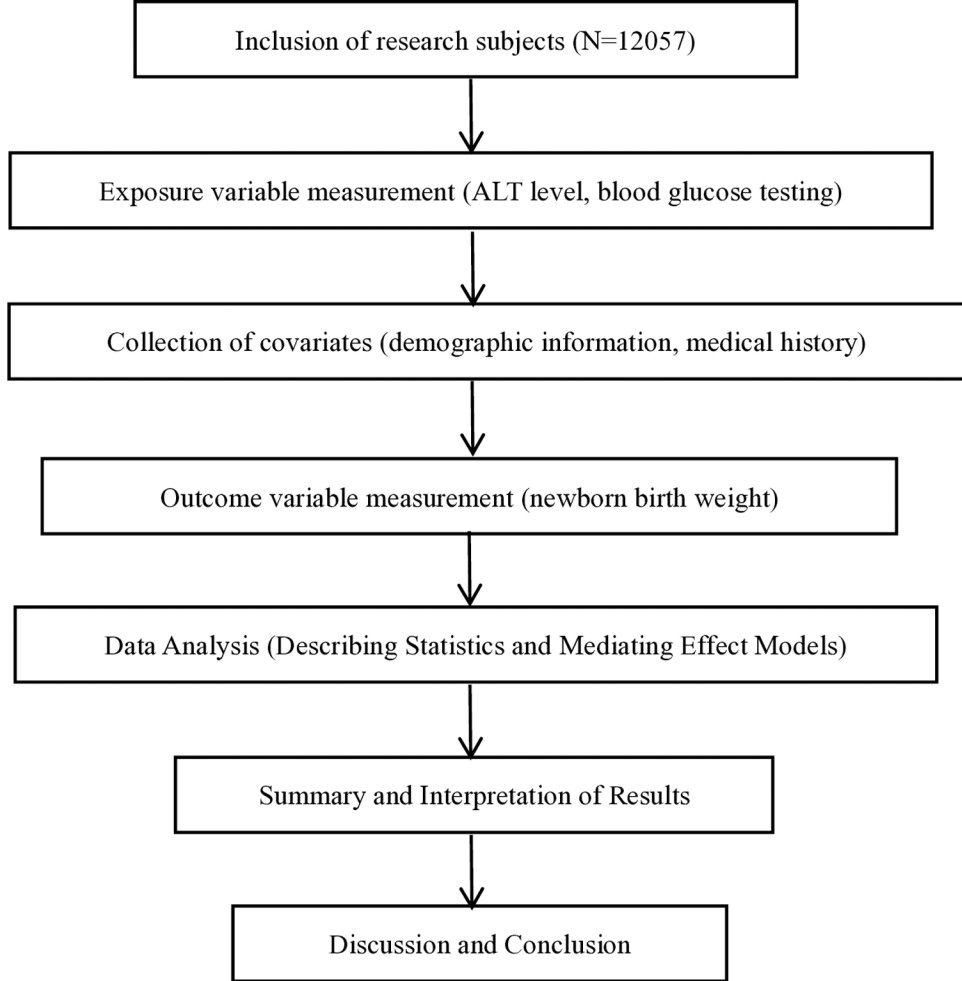

**Fig 1. Research process flowchart.** Explained the research process and key steps of this study.

**Table 1. Comparison of baseline data between non LGA and LGA groups.**

| | Non LGA(n=10783) | LGA(n=1274) | P |
|---|---|---|---|
| age, year | 30.0(6.0) | 30.0(5.0) | <0.001 |
| BMI, kg/m2 | | | |
| <18.5 | 1885(17.6) | 83(6.6) | <0.001 |
| 18.5- | 7977(74.4) | 976(77.3) | |
| 25- | 781(7.3) | 177(14.0) | |
| >30 | 74(0.7) | 26(2.1) | |
| pregnancy weight gain, kg | 13.5(5.0) | 15.0(6.0) | <0.001 |
| nulliparous | 4515(41.0) | 438(34.4) | <0.001 |
| assisted reproduction | 1019(9.5) | 173(13.6) | <0.001 |
| pregnancy hypertension | 166(1.5) | 29(2.3) | 0.049 |
| pre-eclampsia | 167(1.5) | 28(2.2) | 0.082 |
| GDM | 2305(21.4) | 326(25.6) | 0.001 |
| pre-pregnancy diabetes | 115(1.1) | 26(2.0) | 0.002 |
| ICP | 522(4.8) | 72(5.7) | 0.206 |
| thyroid disease | 1097(10.2) | 141(11.1) | 0.320 |
| delivery gestational week, week | 39.0(2.0) | 39.0(1.0) | <0.001 |
| elevated ALT | 675(6.3) | 114(8.9) | <0.001 |

Data are presented as Median (IQR) or number (%).

GDM=Gestational diabetes mellitus, ICP=intrahepatic cholestasis of pregnancy.

### The relationship between early pregnancy ALT levels, the presence of GDM, and whether the infant is LGA

The results of the multiple logistic regression model analysis showed that before adjustment, elevated ALT levels in early pregnancy were significantly positively correlated with LGA (OR: 1.472, 95% CI: 1.196–1.811). After adjusting for age, BMI, pregnancy weight gain, and delivery gestational week, the risk of delivering LGA increased in the elevated ALT group when compared to the normal ALT group (OR: 1.259, 95% CI: 1.011–1.567). After adjusting for age, BMI, pregnancy weight gain, delivery gestational week, nulliparous, assisted reproduction, pregnancy hypertension, pre-pregnancy diabetes, and GDM, the risk of delivering LGA also increased in the elevated ALT group compared to the normal ALT group (OR: 1.248, 95% CI: 1.001–1.556) (Table 2).

Before adjustment, elevated ALT levels were significantly positively correlated with the presence of GDM (OR: 1.698, 95% CI: 1.451–1.987). After adjusting for age, BMI, pregnancy weight gain, and delivery gestational week, the risk of having GDM increased in the elevated ALT group compared to the normal ALT group (OR: 1.553, 95% CI: 1.300–1.856). After

**Table 2. The relationship between ALT levels in early pregnancy and the presence of LGA.**

| LGA | Model 1 | | Model 2 | | Model 3 | |
|---|---|---|---|---|---|---|
| | β(95%CI) | P | β(95%CI) | P | β(95%CI) | P |
| normal ALT | 1.000 | | 1.000 | | 1.000 | |
| elevated ALT | 1.472(1.196,1.811) | <0.001 | 1.259(1.011,1.567) | 0.040 | 1.248(1.001,1.556) | 0.049 |

CI: confidence interval

Model 1: unadjusted model

Model 2: adjusted model, adjusted for age, BMI, pregnancy weight gain, delivery gestational week.

Model 3: adjusted model, adjusted for age, BMI, pregnancy weight gain, delivery gestational week, nulliparous, assisted reproduction, pregnancy hypertension, pre-pregnancy diabetes, GDM.

adjusting for age, BMI, pregnancy weight gain, delivery gestational week, nulliparous, assisted reproduction, pregnancy hypertension, pre-pregnancy diabetes, pre-eclampsia, ICP, and thyroid diseases, the risk of having GDM also increased in the elevated ALT group compared to the normal ALT group (OR: 1.564, 95% CI: 1.306–1.873) (Table 3).

Before adjustment, GDM was significantly positively correlated with LGA (OR: 1.265, 95% CI: 1.106–1.446). After adjusting for age, BMI, pregnancy weight gain, and delivery gestational week, the risk of delivering LGA increased in the GDM group compared to those without GDM (OR: 1.300, 95% CI: 1.125–1.503). After adjusting for age, BMI, pregnancy weight gain, delivery gestational week, nulliparous, assisted reproduction, pregnancy hypertension, pre-pregnancy diabetes, and ALT, the risk of delivering LGA also increased in the GDM group compared to those without GDM (OR: 1.306, 95% CI: 1.129–1.511) (Table 4).

### The mediating role of GDM in the effect of early pregnancy ALT on neonatal birth weight

**Direct effects of ALT.** Before adjustment, ALT had a significant positive direct effect on LGA ($\beta = 0.0402$, 95% CI: 0.0175–0.0623); In Model 2, ALT also had a significant positive direct effect on LGA ($\beta = 0.0257$, 95% CI: 0.0021–0.0531); In Model 3, ALT also had a significant positive direct effect on LGA ($\beta = 0.0291$, 95% CI: 0.0100–0.0635), indicating that an increase in ALT levels during early pregnancy can directly increase the risk of larger than gestational age infants by 0.0291 (Table 5).

**Indirect effects via GDM.** Before adjustment, in the total effect of ALT on LGA, the indirect impact of GDM on LGA was 0.0023 (95% CI: 0.0007–0.0046); In Model 2, GDM also has a significant indirect effect on LGA ($\beta = 0.0023$, 95% CI: 0.0011–0.0054); In Model 3, GDM also had a significant indirect effect on LGA ($\beta = 0.0025$, 95% CI: 0.0012–0.0056) (Table 5).

**Table 3. Relationship between ALT levels in early pregnancy and the presence or absence of GDM.**

| GDM | Model 1 | | Model 2 | | Model 3 | |
|---|---|---|---|---|---|---|
| | β(95%CI) | P | β(95%CI) | P | β(95%CI) | P |
| normal ALT | 1.000 | | 1.000 | | 1.000 | |
| elevated ALT | 1.698(1.451,1.987) | <0.001 | 1.553(1.300,1.856) | <0.001 | 1.564(1.306,1.873) | <0.001 |

CI: confidence interval

Model 1: unadjusted model

Model 2: adjusted model, adjusted for age, BMI, pregnancy weight gain, delivery gestational week.

Model 3: adjusted model, adjusted for age, BMI, pregnancy weight gain, delivery gestational week, nulliparous, assisted reproduction, pregnancy hypertension, pre-eclampsia, ICP, thyroid disease.

**Table 4. Relationship between GDM and LGA.**

| LGA | Model 1 | | Model 2 | | Model 3 | |
|---|---|---|---|---|---|---|
| | β(95%CI) | P | β(95%CI) | P | β(95%CI) | P |
| Non GDM | 1.000 | | 1.000 | | 1.000 | |
| GDM | 1.265(1.106,1.446) | 0.001 | 1.300(1.125,1.503) | <0.001 | 1.306(1.129,1.511) | <0.001 |

CI: confidence interval

Model 1: unadjusted model

Model 2: adjusted model, adjusted for age, BMI, pregnancy weight gain, delivery gestational week.

Model 3: adjusted model, adjusted for age, BMI, pregnancy weight gain, delivery gestational week, nulliparous, assisted reproduction, pregnancy hypertension, pre-pregnancy diabetes, ALT.

**Table 5. Does GDM mediate the indirect effect of ALT levels on LGA.**

|  | Model 1 |  | Model 2 |  | Model 3 |  |
|---|---|---|---|---|---|---|
|  | β(95%CI) | P | β(95%CI) | P | β(95%CI) | P |
| Total effect | 0.0425(0.0203,0.1256) | <0.001 | 0.0280(0.0046,0.0578) | <0.001 | 0.0316(0.0145,0.0672) | <0.001 |
| Direct effect | 0.0402(0.1750,0.0623) | <0.001 | 0.0257(0.0021,0.0531) | <0.001 | 0.0291(0.0100,0.0635) | <0.001 |
| Indirect effect | 0.0023(0.0007,0.0046) | <0.001 | 0.0023(0.0011,0.0054) | <0.001 | 0.0025(0.0012,0.0056) | <0.001 |
| PM(%) | 5.3 | <0.001 | 8.0 | 0.04 | 8.1 | <0.001 |

CI: confidence interval

Model 1: unadjusted model

Model 2: adjusted model, adjusted for age, BMI, pregnancy weight gain, delivery gestational week.

Model 3: adjusted model, adjusted for age, BMI, pregnancy weight gain, delivery gestational week, nulliparous, assisted reproduction, pregnancy hypertension, pre-pregnancy diabetes.

**Proportion of mediation.** Before adjustment, the proportion of indirect effects of GDM to the total effect of ALT on LGA was 5.3% (P<0.05); In Model 2, the indirect effect of GDM accounted for 8.0% (P<0.05) of the total effect of ALT on LGA; In Model 3, the indirect effect of GDM accounted for 8.1% (P<0.05) of the total effect of ALT on LGA (Table 5).

Fig 2 shows the estimated proportions of the association between early pregnancy ALT levels and LGA. The figure displays the estimated values of the indirect effect (IE), direct effect (DE), and the proportion of mediation (IE/(DE+IE)). The mediation model was adjusted for age, BMI, pregnancy weight gain, delivery gestational week, nulliparous, assisted reproduction, pregnancy hypertension, and pre-existing diabetes (Fig 2).

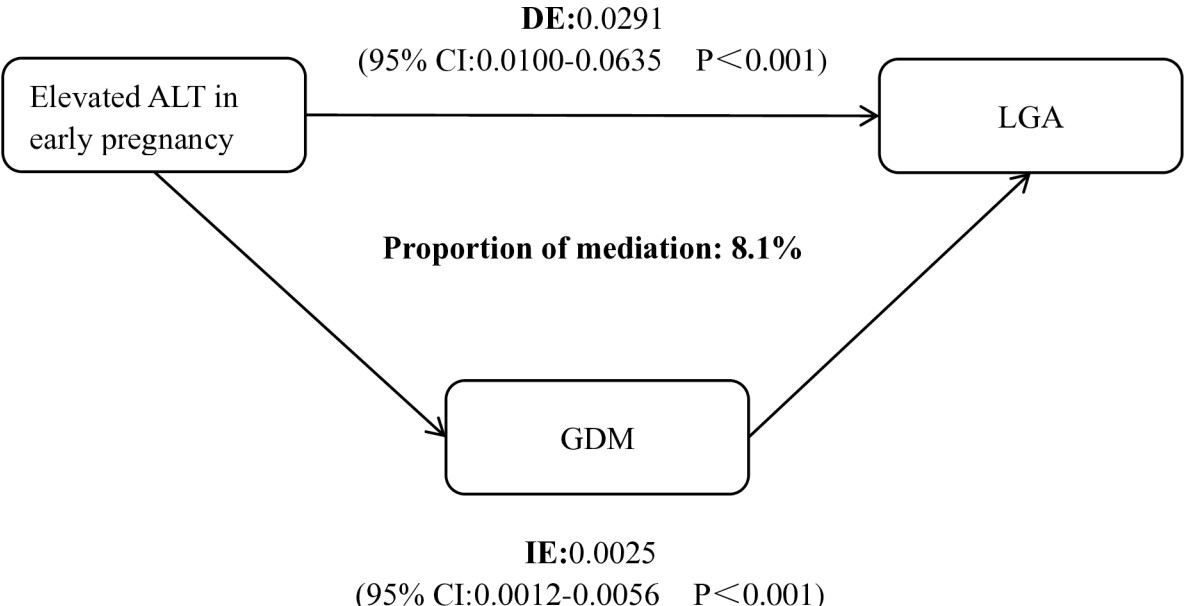

**Fig 2. Estimated proportion of association between early pregnancy ALT levels mediated by GDM and LGA.** IE=indirect effect, DE=direct effect. Adjusted: age, BMI, pregnancy weight gain, delivery gestational week, nulliparous, assisted reproduction, pregnancy hypertension, and pre-existing diabetes.

## Stratified analysis

According to the method of conception, participants were divided into a natural conception group (n = 10,865) and an assisted reproduction group (n = 1,192). Mediation analysis was conducted to examine whether GDM mediated the indirect effect of ALT levels on LGA.

In the natural conception group, after adjusting for age, BMI, pregnancy weight gain, delivery gestational week, nulliparous, assisted reproduction, pregnancy hypertension, and pre-existing diabetes, elevated ALT levels directly affected neonatal birth weight (β = 0.0292, 95% CI: 0.0097–0.0627), while GDM had an indirect effect on neonatal birth weight (β = 0.0026, 95% CI: 0.0010–0.0048), with GDM accounting for 8.4% of the mediation effect.

In the assisted reproduction group, GDM did not mediate the indirect effect of ALT levels on LGA both before adjustment (β = 0.0004, 95% CI: -0.0021–0.0016) and after adjustment (Model 2: β = 0.0007, 95% CI: -0.0009–0.0038; Model 3: β = 0.0007, 95% CI: -0.0019–0.01287) (Table 6).

## Discussion

The main findings of this study are as follows: (1) After adjusting for confounding factors, elevated ALT levels in early pregnancy are associated with an increased likelihood of delivering LGA infants, GDM is associated with a higher rate of LGA births and elevated ALT levels are linked to an increased incidence of GDM. (2) After adjusting for confounding factors, both early pregnancy ALT and GDM influence neonatal birth weight. Elevated ALT levels directly affect neonatal birth weight (β = 0.0291, 95% CI: 0.0100–0.0635), while GDM has an indirect effect (β = 0.0025, 95% CI: 0.0012–0.0056), with GDM accounting for 8.1% of the mediation effect.

The results of this study indicate that elevated ALT levels in early pregnancy are associated with an increased likelihood of delivering LGA infants, consistent with previous findings. A nested case-control study using conditional logistic regression showed that elevated ALT levels in early pregnancy are linked to a fourfold increase in the risk of LGA [3]. A prospective cross-sectional study reported that the prevalence of elevated ALT during pregnancy reached as high as 13.2% [32]. Elevated ALT levels suggest maternal hepatocyte damage [33], leading to insulin resistance in the mother and increase the release of glucose (from gluconeogenesis) and free fatty acids (due to enhanced lipolysis) [34], Maternal glucose is the primary nutrient source for the fetus [35], which may contribute to excessive fetal growth maternal glucose. However,

**Table 6. Stratified analysis by conception method to determine whether GDM mediates the indirect effect of ALT levels on LGA.**

| | | Model 1 | | Model 2 | | Model 3 | |
|---|---|---|---|---|---|---|---|
| | | β(95%CI) | P | β(95%CI) | P | β(95%CI) | P |
| natural conception | Total effect | 0.0429(0.0221,0.0756) | <0.001 | 0.0334(0.0097,0.0627) | 0.04 | 0.0318(0.0123,0.0653) | <0.001 |
| | Direct effect | 0.0406(0.0196,0.0634) | <0.001 | 0.0310(0.0074,0.0612) | 0.04 | 0.0292(0.0097,0.0627) | <0.001 |
| | Indirect effect | 0.0023(0.0008,0.0045) | <0.001 | 0.0024(0.0010,0.0063) | <0.001 | 0.0026(0.0010,0.0048) | <0.001 |
| | PM(%) | 5.2 | <0.001 | 7.3 | 0.04 | 8.4 | <0.001 |
| assisted reproduction | Total effect | 0.0297(-0.0348,0.1128) | 0.48 | -0.0070(-0.0786,0.0623) | 0.88 | 0.0027(-0.0879,0.0937) | 0.96 |
| | Direct effect | 0.0293(-0.0347,0.1115) | 0.44 | -0.0077(-0.0787,0.0683) | 0.92 | 0.0020(-0.0892,0.0965) | 0.96 |
| | Indirect effect | 0.0004(-0.0021,0.0016) | 0.64 | 0.0007(-0.0009,0.0038) | 0.40 | 0.0007(-0.0019,0.01287) | 0.80 |
| | PM(%) | 0.7 | 0.72 | -0.5 | 0.72 | -0.2 | 0.88 |

CI: confidence interval

Model 1: unadjusted model

Model 2: adjusted model, adjusted for age, BMI, pregnancy weight gain, delivery gestational week.

Model 3: adjusted model, adjusted for age, BMI, pregnancy weight gain, delivery gestational week, nulliparous, assisted reproduction, pregnancy hypertension, pre-pregnancy diabetes.

some studies suggest that there is no association between ALT levels during pregnancy and glucose metabolism abnormalities during or after pregnancy [36].

The results of this study show that GDM is associated with an increased likelihood of delivering LGA infants, which is consistent with previous research. A prospective cohort study identified hyperglycemia during pregnancy as a significant predictor of LGA [37]. Another study [38] demonstrated that the placental proteome of women with GDM changes, with ten proteins involved in regulating tissue differentiation and fetal growth, potentially contributing to excessive fetal growth. Additionally, research indicates [39] that maternal insulin resistance in women with GDM leads to increased blood glucose transfer to the fetal circulation, with excess glucose stored as body fat, resulting in fetal overgrowth.

The results of this study indicate that elevated ALT levels in early pregnancy are associated with an increased incidence of GDM, which aligns with some previous findings. A study employing inverse variance weighting combined with two-sample Mendelian randomization (MR) analysis [6] demonstrated a positive causal relationship between ALT and GDM (OR=1.28, 95% CI: 1.05–1.54). Additionally, a prospective cohort study conducted in China that included 17,359 pregnant women [40] suggested that elevated ALT levels within the normal range in early pregnancy can predict the risk of GDM. This may be because increased ALT levels reflect fat accumulation, a hallmark of non-alcoholic fatty liver disease (NAFLD) [41,42], which is closely related to obesity and insulin resistance [43]. An animal model study in mice indicated that hepatic insulin resistance can lead to glucose intolerance, with impaired insulin signaling in the liver resulting in diabetes [44]. However, a nested case-control study [45] involving 256 women who developed GDM found no association between elevated ALT levels and increased GDM risk. The inconsistent conclusions may be attributed to factors such as small sample sizes, differing study designs, variations in liver biomarker detection methods, different diagnostic criteria for GDM, population heterogeneity, and the inclusion of various covariates in adjustment models.

The results of this study indicate that GDM mediates the effect of elevated ALT levels in early pregnancy on neonatal birth weight, acting as a mediator. Previously, no studies have demonstrated that GDM mediates the influence of early pregnancy ALT on LGA. The mediating effect of GDM may be due to its role in inducing insulin resistance in pregnant women; elevated ALT levels also contribute to insulin resistance, exacerbating GDM. This leads to increased release of glucose (from gluconeogenesis) and free fatty acids (due to enhanced lipolysis) in the mother, resulting in greater fetal weight and a higher likelihood of LGA births. In a stratified analysis by mode of conception, GDM showed a mediating effect in the natural conception group, but no mediating effect was observed in the assisted reproduction group. This discrepancy may be attributed to the significantly larger number of participants in the natural conception group compared to the assisted reproduction group.

The strengths of this study lie in the inclusion of all eligible pregnant women, who were managed strictly according to the observation protocol, with no follow-up losses. Currently, there is no literature examining whether GDM mediates the impact of elevated ALT levels in early pregnancy on neonatal birth weight. However, the limitations of this study include its single-center, retrospective design, which may introduce bias, and the inability to fully control for confounding factors influencing pregnancy-related ALT levels and GDM. In future research, we plan to adopt a multicenter prospective design, including participants from different regions and healthcare institutions. This approach will help reduce potential biases arising from geographic and institutional differences, thereby enhancing the external validity and generalizability of the study results. By collecting samples across regions, we will be able to obtain a more diverse representation of the population, thus increasing the reliability of the findings. Furthermore, the prospective design will allow us to better control for confounding factors related to pregnancy-related ALT levels and GDM, further improving the accuracy and scientific rigor of the study.

## Conclusions

Our research suggests a correlation between elevated ALT levels in early pregnancy and increased risk of GDM and LGA, but the causal relationship needs further investigation. Furthermore, GDM partially mediates the effect of elevated ALT

levels on neonatal birth weight. This finding provides scientific and theoretical evidence for reducing the risk of delivering LGA, minimizing adverse pregnancy outcomes, and safeguarding maternal and infant health.

## Supporting information

**S1 Tables. Shapiro-Wilk normality test for continuous variables grouped according to LGA.**
(DOCX)

## Acknowledgments

We would like to acknowledge all the participants of this project and the medical staff for their contribution to this work.

## Author contributions

**Conceptualization:** Wen-Xia Ma, Yu-Bin Ding, Zhaohui Zhong.

**Data curation:** Wen-Xia Ma, Rui Xiao, Yan-Xiao Xiang, Zhaohui Zhong.

**Formal analysis:** Wen-Xia Ma.

**Funding acquisition:** Yu-Bin Ding.

**Investigation:** Zhou Xu, Yan-Xiao Xiang.

**Methodology:** Wen-Xia Ma, Rui Xiao, Yan-Xiao Xiang, Zhaohui Zhong.

**Project administration:** Zhou Xu, Li-Juan Fu, Zhaohui Zhong.

**Resources:** Zhou Xu, Li-Juan Fu.

**Software:** Xiao-Jun Tang.

**Supervision:** Rui Xiao, Xiao-Jun Tang, Yu-Bin Ding.

**Validation:** Wen-Xia Ma, Rui Xiao, Xiao-Jun Tang.

**Visualization:** Wen-Xia Ma, Xiao-Jun Tang, Yu-Bin Ding.

**Writing – original draft:** Wen-Xia Ma, Shao-Min Yu.

**Writing – review & editing:** Wen-Xia Ma, Shao-Min Yu.

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
