## [Decision Letter · Decision Letter 0]

2 Jan 2025

PONE-D-24-42399The effect of early pregnancy ALT elevation on neonatal birth weight: the mediating role of gestational diabetes mellitusPLOS ONE

Dear Dr. Zhong,

Thank you for submitting your manuscript to PLOS ONE. After careful consideration, we feel that it has merit but does not fully meet PLOS ONE’s publication criteria as it currently stands. Therefore, we invite you to submit a revised version of the manuscript that addresses the points raised during the review process.

We look forward to receiving your revised manuscript.

Kind regards,

Fumihiko Namba

Academic Editor

PLOS ONE

“This work was supported by the National Key Research and Development Program of China (no: 2023YFC2705900) and the Open Fund of Chongqing Maternal and Child Disease Control and Public Health Research Center (no: CQFYSJ01001).”

3. In this instance it seems there may be acceptable restrictions in place that prevent the public sharing of your minimal data. However, in line with our goal of ensuring long-term data availability to all interested researchers, PLOS’ Data Policy states that authors cannot be the sole named individuals responsible for ensuring data access (http://journals.plos.org/plosone/s/data-availability#loc-acceptable-data-sharing-methods).

6. Please include a separate caption for each figure in your manuscript.

7. Please include your tables as part of your main manuscript and remove the individual files. Please note that supplementary tables (should remain/ be uploaded) as separate "supporting information" files.

Reviewers' comments:

Reviewer's Responses to Questions

**Comments to the Author**

1. Is the manuscript technically sound, and do the data support the conclusions?

Reviewer #1: Yes

Reviewer #2: Partly

2. Has the statistical analysis been performed appropriately and rigorously? 

Reviewer #1: Yes

Reviewer #2: Yes

3. Have the authors made all data underlying the findings in their manuscript fully available?

Reviewer #1: Yes

Reviewer #2: Yes

4. Is the manuscript presented in an intelligible fashion and written in standard English?

Reviewer #1: Yes

Reviewer #2: Yes

5. Review Comments to the Author

Reviewer #1: The study addresses an important knowledge gap regarding the mediating role of GDM in the relationship between early pregnancy ALT levels and neonatal birth weight. The methodology is sound, with: Appropriate study design (retrospective cohort); Large sample size (12,057 patients); Clear inclusion criteria; Proper statistical analyses including mediation modeling

Adequate control for confounding factors. The findings are clinically relevant and could impact prenatal care practices by suggesting that monitoring and managing ALT levels in early pregnancy might help reduce GDM and LGA incidence.

I suggest that results section would benefit from clearer organization of the mediation analysis findings. Methods Section could benefit from more detail about the mediation analysis methodology and should clarify the power calculation or sample size determination. The methods section lacks information about the methods utilized to evaluate ALT and glucose levels; the cut-off values and their coefficient of variability need to be specified in detail for reproducibility.

In the statistical analysis part the authors report that they have "Continuous variables between groups were compared using the Mann-Whitney U test and analysis of variance (ANOVA), while categorical variables between groups were compared using the Chi-Square Test." Mann-Whitney U test is used for non-parametric variables and should be given by median and IQR or min-max. If the values are given by mean and SD the variables must be normally distributed and the diffrence for the means are calculated by Student's t test. Please give information about the data distribution.

Reviewer #2: Title and abstract

1. In your opinion, how does the title and abstract effectively capture the content and focus of the manuscript? Please suggest any improvement if needed.

Title: Write properly

Abstract: Increase the contents of research significance of this study

In conclusion part of the abstract, please not only repeated the results, but should focus on and advise medical providers to increase their policy measures on the research results.

2. Did you find that the background and information provided was sufficient to understand the research? Would you have any other suggestions?

I would rather like to see some background statistics on the prevalence of ALT in China or the prevalence of the study, to give the reader some background information about the research background. I think this introduction requires more review of the literature.

Materials and Methods

3.Based on your expertise, how will you assess the clarity of the methods used in your study?

What are the tests for exposure variables?

Why did this study take the diagnosis of blood glucose at 13 weeks rather than using the criteria for gestational diabetes, It suggested to add literature here, how do you control all the confounding factors in the study? Each variable will need to be supported by the literature. Should the covariates also include pregnant women with dyslipidemia?

How is the sample size calculated?

When was your informed consent form signed? How to sign it?

Statistical analysis of the mediation effect model needs more literature to support.

4. If applicable, please share your views on the novelty of the study findings.

The results are very well presented

discuss

5. Are the findings described by the authors relevant to the results? From your perspective, are the results related to the objectives of the study and the broader research context?

The results are well discussed and suggest satisfactory supportive reasoning, similar studies are supported and the authors try to find supportive studies in each significant variable, but I still needed that the introduction should support all the issues discussed.

6. In your opinion, how do these conclusions fit with the research results?

This manuscript is written clearly and all limitations are highlighted, but in this study it is a single-center, retrospective design that may introduce bias. Insufficient control of confounding factors to fully control the confounding factors affecting pregnancy-related ALT levels and GDM, multicenter studies can increase the sample diversity and reduce the bias caused by factors such as geography, making the results more universal.

Charts and tables

7. If the author provides the charts, are they clear and clearly visible? Are these numbers unnecessarily modified?

Column a diagram of the study steps and the intervention for each step so that the reader understands the entire study process.

8. Is this statistical analysis suitable for this study?

Yes

9. Are these references relevant to this study?

Yes

10. Is the referenced style correct?

Yes

11. Are you concerned about the similarity with other articles published by the same author or any other questions?

no-one

Competitive interests

12. Did any authors competing interests cause concern about the validity of the study, namely, did the authors competing interests generate competing interests in the reporting of the results and conclusions?

not have

6. PLOS authors have the option to publish the peer review history of their article (what does this mean? ). If published, this will include your full peer review and any attached files.

**Do you want your identity to be public for this peer review?** For information about this choice, including consent withdrawal, please see our Privacy Policy .

Reviewer #1: **Yes: ** A.Seval Ozgu-Erdinc

Reviewer #2: No

---

## [Author Response · Author response to Decision Letter 0]

16 Jan 2025

Dear Editor and Reviewers,

Thank you for the opportunity to revise our manuscript titled "The Effect of Early Pregnancy ALT Elevation on Neonatal Birth Weight: The Mediating Role of Gestational Diabetes Mellitus" (manuscript number: PONE-D-24-42399). We are truly grateful for the insightful and constructive feedback provided by the reviewers, which has significantly contributed to enhancing the quality and clarity of our work.

We would also like to make a humble request regarding authorship. Dr. Zhou Xu, who was the data owner at Sichuan Jinxin Xinan Women and Children’s Hospital and made substantial contributions to manuscript design, drafting, and revision, was unintentionally omitted as a co-first author. We deeply regret this oversight and would like to amend the manuscript to reflect his rightful contribution. Similarly, Prof. Yu-Bin Ding should be designated as a co-corresponding author, as his contributions are integral to the study. After reassessing the contributions, we concluded that both Dr. Zhou Xu and Prof. Yu-Bin Ding's roles in the research are of equal significance. All authors have been consulted, and we have unanimously agreed to make this adjustment.

Attached, please find our point-by-point responses to each reviewer comment. We have carefully addressed all suggestions and revisions, and we hope that the revised manuscript meets the journal’s standards for publication.

We welcome any further comments and sincerely appreciate the time and effort the reviewers have dedicated to evaluating our work. If additional documentation or information is required for the authorship change, please do not hesitate to let us know.

Thank you again for your consideration and support.

Sincerely,

Zhao-Hui Zhong, Prof.

Department of Obstetrics and Gynecology, Women and Children’s Hospital of Chongqing Medical University

Joint International Research Laboratory of Reproduction and Development of the Ministry of Education of China

School of Public Health, Chongqing Medical University, Chongqing 400016, China

Point-to-point response to reviewers and editors

Editors #

1.Please ensure that your manuscript meets PLOS ONE's style requirements, including those for file naming.

Response:

Thank you for your feedback and guidance. We have ensured that the manuscript adheres to PLOS ONE’s formatting requirements, using the provided templates and adjusting the title, author information, and file naming accordingly.

2.Thank you for stating the following financial disclosure:

“This work was supported by the National Key Research and Development Program of China (no: 2023YFC2705900) and the Open Fund of Chongqing Maternal and Child Disease Control and Public Health Research Center (no: CQFYSJ01001).”

Response:

Thank you for your feedback regarding the financial disclosure statement. We have revised the statement as follows:

“This work was supported by the Open Fund of Chongqing Maternal and Child Disease Control and Public Health Research Center (no: CQFYSJ01001). The funders had no role in study design, data collection and analysis, decision to publish, or preparation of the manuscript.”

Please update the online submission form accordingly.

3.In this instance it seems there may be acceptable restrictions in place that prevent the public sharing of your minimal data. However, in line with our goal of ensuring long-term data availability to all interested researchers, PLOS’ Data Policy states that authors cannot be the sole named individuals responsible for ensuring data access

(http://journals.plos.org/plosone/s/data-availability#loc-acceptable-data-sharing-methods).

Response:

Thank you for your valuable feedback and guidance. We fully understand the importance of ensuring long-term data availability in compliance with the journal's data policy. Unfortunately, we do not have non-author contact information (phone/email/hyperlink) available for direct data access requests. To address this, we have identified institutional representatives who were not involved in the study and are not listed as authors on the manuscript, but who can hold the data and respond to external data access requests. The contact information is as follows: email: pinyi.chen@cqmu.edu.cn; Phone: +86 23 68485111; Fax: +86 23 63846904].

4.Please amend either the title on the online submission form (via Edit Submission) or the title in the manuscript so that they are identical.

Response:

Thank you for your comment. We have made the necessary amendment to ensure that the title in the manuscript exactly matches the one on the online submission form, as requested.

Response:

Thank you for your feedback and guidance. We have moved the ethics statement to the methods section and removed the ethics statements in other sections.

6. Please include a separate caption for each figure in your manuscript.

Response:

Thank you for your feedback and guidance. Each figure now has a separate, clearly written caption.

7.Please include your tables as part of your main manuscript and remove the individual files. Please note that supplementary tables (should remain/ be uploaded) as separate "supporting information" files.

Response:

Thank you for your feedback and guidance. All tables have been incorporated into the main manuscript, and supplementary tables have been uploaded as separate supporting information files.

Reviewer 1#

Comment 1:

The results section would benefit from clearer organization of the mediation analysis findings.

Response:

Thank you for your valuable suggestion. In response, we have reorganized the mediation analysis findings in the results section to improve clarity. Specifically, we have added subheadings to differentiate between "Direct Effects of ALT," "Indirect Effects via GDM," and "Proportion of Mediation." Each section now presents the mediation effects in a consistent format, including β values, confidence intervals, and percentages of mediation for both the natural conception and assisted reproduction groups. This restructuring should provide a clearer understanding of the results. Please refer to the revised tables and the corresponding description of results.

Comment 2:

Methods section could benefit from more detail about the mediation analysis methodology.

Response:

We appreciate this feedback. To address this, we will: Provide additional details on the mediation analysis methodology, including the statistical model used (e.g., causal steps approach, bootstrapping, or structural equation modeling). Include a description of the software and packages used for mediation analysis.

Please refer to the revised

Comment 3:

The Methods section should clarify the power calculation or sample size determination.

Response:

Thank you for the valuable feedback from the reviewer. Regarding the calculation of sample size, we estimated it based on logistic regression analysis and mediation analysis, and ultimately determined that the minimum sample size required for this study is 412.In practice, we included 12,057 research subjects, which fully satisfies the sample size requirement. We have added a section on "Sample Size Calculation" in the Methods section.

Comment 4:

The methods section lacks information about the methods utilized to evaluate ALT and glucose levels; the cut-off values and their coefficient of variability need to be specified in detail for reproducibility.

Response:

Thank you for your helpful comment. In response, we have now included detailed information on the laboratory methods used for measuring ALT and glucose levels in the Methods section. Specifically, we have provided the cut-off values and the coefficient of variability for these assays to ensure reproducibility of our findings. This additional information should help clarify the methodology used for these measurements.

Comment 5:

Mann-Whitney U test is used for non-parametric variables and should be given by median and IQR or min-max. If the values are given by mean and SD, the variables must be normally distributed, and differences for the means are calculated by Student's t-test. Please give information about the data distribution.

Response:

Thank you for the valuable feedback. We have performed normality tests (Shapiro-Wilk) for all continuous variables. For non-normally distributed variables, we reported medians and IQRs and used the Mann-Whitney U test. For normally distributed variables, we used means ± SD and applied t-tests.

These changes are now reflected in the Statistical Analysis section and Table 1. A supplementary table has been added to the results section, displaying the Shapiro Wilk normality test for each variable.

Reviewer 2#

Comment 1:

In conclusion part of the abstract, please not only repeated the results, but should focus on and advise medical providers to increase their policy measures on the research results.

Response:

Thank you for the constructive comments from the reviewer. We have added policy recommendations for healthcare providers in the conclusion section of the abstract, which not only emphasizes the value of the research but also promotes its practical application.

Comment 2:

I would rather like to see some background statistics on the prevalence of ALT in China or the prevalence of the study, to give the reader some background information about the research background. I think this introduction requires more review of the literature.

Response:

Thank you for your valuable suggestion. We have now incorporated relevant data on the prevalence of ALT abnormalities in China and referenced studies that highlight the significance of ALT levels in pregnancy.

Comment 3:

What are the tests for exposure variables?

Response:

Thank you for your question. The exposure variable, elevated ALT levels, was assessed using enzymatic methods, following established clinical laboratory protocols. We have now clarified this in the Methods section of the manuscript to provide a more detailed explanation of the testing procedure.

Comment 4:

Why did this study take the diagnosis of blood glucose at 13 weeks rather than using the criteria for gestational diabetes, It suggested to add literature here.

Response:

Thank you for your valuable feedback. In this study, all patients were screened for GDM in strict accordance with the Diagnosis and Treatment Guidelines for Gestational Hyperglycemia (2022). At 24-28 weeks of pregnancy, a 75 g oral glucose tolerance test (OGTT) was performed: the blood glucose thresholds on an empty stomach and 1 hour and 2 hours after oral glucose were 5.1, 10.0, and 8.5 mmol/L, respectively. If the blood glucose value reached or exceeded the specified criteria at any time point, it was diagnosed as GDM.

The aim of this study is to explore whether gestational diabetes mellitus (GDM) acts as a mediator in the relationship between early pregnancy ALT levels and neonatal birth weight. Since ALT levels were measured at 11-13 weeks of pregnancy, we wanted to emphasize that the GDM screening was conducted after the ALT measurement. This is why we stated that blood glucose was tested after 13 weeks. We apologize for any confusion caused by the lack of clarity in our original expression and have made the necessary revisions in the manuscript.

Comment 5:

How do you control all the confounding factors in the study? Each variable will need to be supported by the literature. Should the covariates also include pregnant women with dyslipidemia?

Response:

Thank you for your insightful comment regarding the control of confounding factors in the study. We recognize the importance of controlling for potential confounders to ensure the validity of our results.

In this study, the following methods were used to effectively control confounding factors and ensure the accuracy and reliability of the results:

1.Selection of Relevant Covariates: Based on the literature, multiple variables that could influence pregnancy outcomes (such as maternal age, pre-pregnancy BMI, gestational weight gain, gestational hypertension, thyroid disorders, etc.) were selected as covariates to control for potential confounding effects.

2.Multivariable Regression Analysis: A multivariable regression analysis was used to simultaneously account for multiple potential factors that could influence the outcome, effectively controlling for confounding factors and ensuring a more accurate assessment of the relationship between ALT, GDM, and LGA.

3.Mediation Analysis: Mediation analysis was conducted to explore whether GDM acts as a mediator in the relationship between elevated ALT and LGA, controlling for both direct and indirect effects, and revealing the complex causal relationships.

4.Stratified Analysis: Stratified Analysis can be used to further refine the results and ensure accurate estimation of whether GDM plays a mediating role in the relationship between ALT elevation and LGA in different subgroups.

We controlled for the following factors:

1.Maternal age:

Maternal age is a well-established factor influencing pregnancy outcomes. Studies have shown that advanced maternal age is associated with an increased risk of GDM and larger newborns, and it may also affect ALT levels.

Reference:

Londero AP, Rossetti E, Pittini C, et al. Maternal age and the risk of adverse pregnancy outcomes: a retrospective cohort study. BMC PREGNANCY CHILDB. [Journal Article]. 2019 2019-7-23;19(1):261. doi: 10.1186/s12884-019-2400-x

Kim EH, Lee J, Lee SA, et al. Impact of Maternal Age on Singleton Pregnancy Outcomes in Primiparous Women in South Korea. J CLIN MED. [Journal Article]. 2022 2022-2-12;11(4). doi: 10.3390/jcm11040969

2.Pre-pregnancy BMI:

Pre-pregnancy BMI has been consistently identified as a key factor influencing gestational weight gain, GDM development, and neonatal birth weight. Higher pre-pregnancy BMI is associated with abnormal liver function during pregnancy, as obesity may lead to liver fat deposition and metabolic abnormalities.

Reference:

Liu P, Xu L, Wang Y, et al. Association between perinatal outcomes and maternal pre-pregnancy body mass index. OBES REV. [Journal Article; Meta-Analysis; Review]. 2016 2016-11-1;17(11):1091-102. doi: 10.1111/obr.12455

Li M, Wang L, Du Z, et al. Joint effect of maternal pre-pregnancy body mass index and folic acid supplements on gestational diabetes mellitus risk: a prospective cohort study. BMC PREGNANCY CHILDB. [Journal Article]. 2023 2023-3-23;23(1):202. doi: 10.1186/s12884-023-05510-y

3.Gestational Weight Gain:

Excessive weight gain during pregnancy is associated with in

---

## [Decision Letter · Decision Letter 1]

11 Mar 2025

PONE-D-24-42399R1The effect of early pregnancy ALT elevation on neonatal birth weight: the mediating role of gestational diabetes mellitusPLOS ONE

Dear Dr. Zhong,

Thank you for submitting your manuscript to PLOS ONE. After careful consideration, we feel that it has merit but does not fully meet PLOS ONE’s publication criteria as it currently stands. Therefore, we invite you to submit a revised version of the manuscript that addresses the points raised during the review process.

We look forward to receiving your revised manuscript.

Kind regards,

Fumihiko Namba

Academic Editor

PLOS ONE

**Journal Requirements:**

Reviewers' comments:

Reviewer's Responses to Questions

**Comments to the Author**

1. If the authors have adequately addressed your comments raised in a previous round of review and you feel that this manuscript is now acceptable for publication, you may indicate that here to bypass the “Comments to the Author” section, enter your conflict of interest statement in the “Confidential to Editor” section, and submit your "Accept" recommendation.

Reviewer #1: All comments have been addressed

Reviewer #3: All comments have been addressed

2. Is the manuscript technically sound, and do the data support the conclusions?

Reviewer #1: Yes

Reviewer #3: Partly

3. Has the statistical analysis been performed appropriately and rigorously? 

Reviewer #1: Yes

Reviewer #3: No

4. Have the authors made all data underlying the findings in their manuscript fully available?

Reviewer #1: Yes

Reviewer #3: Yes

5. Is the manuscript presented in an intelligible fashion and written in standard English?

Reviewer #1: Yes

Reviewer #3: No

6. Review Comments to the Author

**Reviewer #1:**  (No Response)

**Reviewer #3:**  Dear Editor

Thank you very much for the opportunity to review this article. The previous reviewers' comments are very valuable and almost all the important points have been made, but a few important points are worth considering.

1- I check the manuscript and found that near to all parts of discussion, abstracts and other parts of the manuscript is written by AI. Is it in line with the journal policies?

2- The method section is incompletely written. The inclusion and exclusion criteria are not mentioned.

3- In the results section, Table 1, two very similar numbers in maternal age and age at delivery have significant value. Please check the data and if possible, request the authors to send the data sheet.

4- In conclusion part how the authors conclude “that reducing ALT levels in early pregnancy can help decrease the incidence of GDM as well as the likelihood of delivering LGA infants. “Based on the method and results this conclusion may not be appropriate.

7. PLOS authors have the option to publish the peer review history of their article (what does this mean? ). If published, this will include your full peer review and any attached files.

**Do you want your identity to be public for this peer review?** For information about this choice, including consent withdrawal, please see our Privacy Policy .

Reviewer #1: **Yes: ** A.Seval Ozgu-Erdinc

Reviewer #3: No

---

## [Author Response · Author response to Decision Letter 1]

13 Mar 2025

Dear Editor and Reviewers,

Thank you for the opportunity to revise our manuscript titled "The Effect of Early Pregnancy ALT Elevation on Neonatal Birth Weight: The Mediating Role of Gestational Diabetes Mellitus" (manuscript number: PONE-D-24-42399R1). We are truly grateful for the insightful and constructive feedback provided by the reviewers, which has significantly contributed to enhancing the quality and clarity of our work.

Attached, please find our point-by-point responses to each reviewer comment. We have carefully addressed all suggestions and revisions, and we hope that the revised manuscript meets the journal’s standards for publication.

We welcome any further comments and sincerely appreciate the time and effort the reviewers have dedicated to evaluating our work.

Thank you again for your consideration and support.

Sincerely,

Zhao-Hui Zhong, Prof.

Department of Obstetrics and Gynecology, Women and Children’s Hospital of Chongqing Medical University

Joint International Research Laboratory of Reproduction and Development of the Ministry of Education of China

School of Public Health, Chongqing Medical University, Chongqing 400016, China

Point-to-point response to reviewers and editors

Editors #

1.Please review your reference list to ensure that it is complete and correct. If you have cited papers that have been retracted, please include the rationale for doing so in the manuscript text, or remove these references and replace them with relevant current references. Any changes to the reference list should be mentioned in the rebuttal letter that accompanies your revised manuscript. If you need to cite a retracted article, indicate the article’s retracted status in the References list and also include a citation and full reference for the retraction notice.

Response:

Thank you for your feedback. We have carefully reviewed all references cited in our manuscript and verified their status using PubMed and other relevant databases. We confirm that none of the references have been retracted. No changes were required in the reference list, but we appreciate the opportunity to ensure its accuracy. In response to the reviewer's comments, we have added an additional reference (Reference 31), which has also not been retracted, to further support our findings.

Reviewer 1#

Response:

We appreciate that Reviewer #1 considers all previous comments to have been adequately addressed. Thank you for your time and constructive feedback.

Reviewer 3#

Comment 1:

I check the manuscript and found that near to all parts of discussion, abstracts and other parts of the manuscript is written by AI. Is it in line with the journal policies?

Response:

Thank you for your attention to the language quality of our manuscript and for your important reminder regarding the use of AI tools. We fully understand the academic community's cautious attitude toward AI-generated content and would like to make a formal clarification: The design of this study, data analysis, extraction of key conclusions, and interpretation of results were all independently completed by our research team, without the use of any AI tools to generate text content directly. In order to improve the accuracy of language expression, we only use Grammarly for language refinement in the final revision stage, which includes correcting grammar errors, optimizing sentence structure, and standardizing terminology. Grammarly is a widely accepted tool for grammar and language enhancement and does not generate content. All refinements were based on the original text provided by the authors, ensuring that the core academic content remained unaffected.

To further demonstrate the originality of our manuscript, we can provide relevant materials from the research process, including the original code used for data analysis and multiple draft revisions of the manuscript. These materials clearly illustrate the development and creation process of the paper.

Partial original code for data analysis:

Model 1

help(package="mediation")

library(mediation)

b <- lm(gdm~alt1, data=zong)

c <- lm(lga~alt1+gdm, data=zong)

contcont <- mediate(b, c, sims=50, treat="alt1", mediator="gdm")

summary(contcont)

plot(contcont)

Model 2

library(mediation)

b <- lm(gdm~alt1+age+bmi3+weightgain+gwod, data=zong)

c <- lm(lga~alt1+gdm+age+bmi3+weightgain+gwod, data=zong)

contcont <- mediate(b, c, sims=50, treat="alt1", mediator="gdm")

summary(contcont)

plot(contcont)

Model 3

library(mediation)

b <- lm(gdm~alt1+age+bmi3+weightgain+gwod+g1+mc+pih+qiangdm, data=zong)

c <- lm(lga~alt1+gdm+age+bmi3+weightgain+gwod+g1+mc+pih+qiangdm, data=zong)

contcont <- mediate(b, c, sims=50, treat="alt1", mediator="gdm")

summary(contcont)

plot(contcont)

Multiple version revision records of paper writing:

The first draft of the manuscript was written in Chinese, starting on September 12, 2024, and completed on September 17, 2024. The second draft, also in Chinese, was started on September 19, 2024, with the addition of stratified analysis in the results section and completed on September 22, 2024. The third draft was written in English, translating the previous version, starting on September 22, 2024, and completed on September 23, 2024.

Once again, I sincerely appreciate the reviewer’s valuable feedback. If there are still any doubts regarding certain parts of the manuscript, I am more than happy to provide additional details or explanations to clear up any misunderstandings.

Comment 2:

The method section is incompletely written. The inclusion and exclusion criteria are not mentioned.

Response:

Thank you for pointing this out. We have now explicitly stated the inclusion and exclusion criteria in the methods section.

Inclusion Criteria: single live birth patients; patients who underwent liver function tests in early pregnancy (11-13 weeks); patients who underwent a 75g oral glucose tolerance test (OGTT) between 24-28 weeks of pregnancy.

Exclusion criteria: patients with unknown gestational age or birth weight; patients with multiple pregnancies; patients with viral hepatitis.

Comment 3:

In the results section, Table 1, two very similar numbers in maternal age and age at delivery have significant value. Please check the data and if possible, request the authors to send the data sheet.

Response:

Thank you for your careful review and valuable feedback. We have rechecked the data for maternal age and age at delivery in Table 1.

In our study, maternal age and delivery gestational week were analyzed as continuous variables. After performing normality tests, we found that these two variables were not normally distributed, so we used the median and interquartile range for description and conducted a Mann-Whitney U test for group comparisons.Therefore, the similar numbers in Table 1 between the two groups represent the median (interquartile range). However, due to the large sample size and the different distributions of these two variables between the groups, a significant statistical difference was observed between the groups.

To ensure accuracy, we have thoroughly reviewed our dataset and confirmed that the values are correct. Additionally, we have referenced relevant literature in the manuscript to provide further clarification. If necessary, we are happy to provide additional details on the data processing steps. Please let us know if further clarification is needed.

Reference:

Hart A. Mann-Whitney test is not just a test of medians: differences in spread can be important. BMJ-BRIT MED J. [Comparative Study; Journal Article]. 2001 2001-8-18;323(7309):391-3. doi: 10.1136/bmj.323.7309.391

Comment 4:

In conclusion part how the authors conclude “that reducing ALT levels in early pregnancy can help decrease the incidence of GDM as well as the likelihood of delivering LGA infants. “Based on the method and results this conclusion may not be appropriate.

Response:

Thank you for your insightful comment regarding the conclusion. We agree that the direct conclusion that "reducing ALT levels in early pregnancy can help decrease the incidence of GDM as well as the likelihood of delivering LGA infants" may not be fully supported by our current data. While our results suggest a potential association between elevated ALT levels in early pregnancy and an increased risk of GDM and LGA infants, we acknowledge that causality cannot be firmly established due to the study's observational nature. Therefore, we will revise the conclusion to more accurately reflect the study design and findings, and provide a more cautious interpretation of the results. We will revise the manuscript accordingly to clarify this point.

---

## [Decision Letter · Decision Letter 2]

24 Mar 2025

The Effect of Early Pregnancy ALT Elevation on Neonatal Birth Weight: The Mediating Role of Gestational Diabetes Mellitus

PONE-D-24-42399R2

Dear Dr. Zhong,

We’re pleased to inform you that your manuscript has been judged scientifically suitable for publication and will be formally accepted for publication once it meets all outstanding technical requirements.

Kind regards,

Fumihiko Namba

Academic Editor

PLOS ONE

Reviewers' comments:

Reviewer's Responses to Questions

**Comments to the Author**

1. If the authors have adequately addressed your comments raised in a previous round of review and you feel that this manuscript is now acceptable for publication, you may indicate that here to bypass the “Comments to the Author” section, enter your conflict of interest statement in the “Confidential to Editor” section, and submit your "Accept" recommendation.

Reviewer #3: All comments have been addressed

2. Is the manuscript technically sound, and do the data support the conclusions?

Reviewer #3: Yes

3. Has the statistical analysis been performed appropriately and rigorously? 

Reviewer #3: Yes

4. Have the authors made all data underlying the findings in their manuscript fully available?

Reviewer #3: Yes

5. Is the manuscript presented in an intelligible fashion and written in standard English?

Reviewer #3: Yes

6. Review Comments to the Author

Reviewer #3: Thank you, This is a valuable study. The authors responded all comments properly and no need for further comments.

7. PLOS authors have the option to publish the peer review history of their article (what does this mean? ). If published, this will include your full peer review and any attached files.

**Do you want your identity to be public for this peer review?** For information about this choice, including consent withdrawal, please see our Privacy Policy .

Reviewer #3: **Yes: ** Marjan Ghaemi

---

## [Editor Report · Acceptance letter]

PONE-D-24-42399R2

PLOS ONE

Dear Dr. Zhong,

I'm pleased to inform you that your manuscript has been deemed suitable for publication in PLOS ONE. Congratulations! Your manuscript is now being handed over to our production team.

Kind regards,

on behalf of

Dr. Fumihiko Namba

Academic Editor

PLOS ONE